



# Comparison of the aerosol optical properties and size distribution retrieved by Sun photometer with in-situ measurements at mid-latitude.

Chauvigné Aurélien[1], Sellegri Karine[1], Hervo Maxime[1,2], Montoux Nadège[1], Freville Patrick[1], Goloub Philippe[3].

[1] Laboratoire de Météorologie Physique, OPGC, CNRS, Université Blaise Pascal, BP 10448, 63000 Clermont-Ferrand, France.
[2] Federal Office of Meteorology and Climatology MeteoSwiss, Payerne, Switzerland.
[3] Laboratoire d'Optique Atmosphérique, Université des Sciences et Techniques de Lille, Villeneuve d'Ascq, France

*Correspondence to:* Karine Sellegri (K.Sellegri@opgc.univ-bpclermont.fr)

Aerosols influence the Earth radiative budget through scattering and absorption of solar radiation. Several methods are used to investigate aerosol properties and thus quantify their direct and indirect impacts on climate. At the Puy de Dôme station, continuous high altitude near surface *in-situ* measurements and low altitude ground-based remote sensing atmospheric column measurements give the opportunity to compare the aerosol extinction measured with both methods over a one year period. To our knowledge, it is the first time that such a comparison is realized with continuous measurements of a high altitude site during a long term period. This comparison addresses to which extend near surface *in-situ* measurements are representative of the whole atmospheric column, the aerosol Mixing Layer (ML), or the Free Troposphere (FT). In particular, the impact of multi aerosol layers events detected using LIDAR backscatter profiles is analysed. A good correlation between *in-situ* aerosol extinction coefficient and Aerosol Optical Depth (AOD) measured by the Aerosol Robotic Network (AERONET) Sun photometer is observed with a correlation coefficient around 0.80, indicating that the *in-situ* measurements station is representative of the overall atmospheric column. After filtering for multilayer cases and correcting for each layer optical contribution (ML and FT), the atmospheric structure seems to be the main factor influencing the comparison between the two measurement techniques. When the site lies in the ML, the *in-situ* extinction represents 45% of the Sun photometer ML extinction while when the site lies within the FT, the *in-situ* extinction is more than two times higher than the FT Sun photometer extinction. Remote sensing retrievals of the aerosol particle size distributions (PSD) from the Sun photometer observations are then compared to the near surface *in-situ* measurements, at dry and at ambient relative humidities. When *in-situ* measurements are considered at dry state, the *in-situ* fine mode diameters are 44% higher than the Sun photometer-retrieved diameters and *in-situ* volume concentrations are 20% lower than of the Sun photometer-retrieved fine mode concentration. Using a parametrised hygroscopic growth factor applied to aerosol diameters, the difference between *in-situ* and retrieved diameters grows larger. Coarse mode *in-situ* diameter and concentrations show a good correlation with retrieved particle size distributions from remote sensing.



## 1. Introduction

Over the last decade, aerosol studies have increased significantly due to the large uncertainty associated to their impact on climate in global models (IPCC, 2014). The important variability of aerosol size, concentration and composition partly drive this uncertainty on their direct and indirect radiative effects (Nemesure et al., 1995; Boucher and Anderson, 1995; Pilinis and Li, 1998). Boucher and Anderson (1995) have calculated that aerosol radiative forcing is more sensitive to small particles (geometric volume mean diameter less than 0.2μm) and that variations on particle size and composition can influence the aerosol direct radiative forcing by around 20%. Another important parameter that is not well included in radiative models is the humidity effect on aerosol properties. Pilinis et al. (1995) have estimated from a box model an increase of the aerosol radiative forcing by a factor of 2.1 for a relative humidity varying from 40 to 80%. Hervo et al. (2014) have shown important hygroscopic enhancement of aerosol optical properties at Puy de Dôme station (PUY) according different air mass origins over a two years period.

Both aerosol horizontal and vertical distributions determine the magnitude of the direct and indirect radiative effects and is still a limit for general aerosol studies (Laj et al., 2009). The aerosol concentration is typically higher within the low tropospheric layer (Mixing Layer (ML)) than in the Free Troposphere (FT), but aerosol layers such as desert dust, marine aerosol or volcanic ash can also be transported at high altitude above the ML over large distances. Hence, ground-based measurements are not always representative of the whole atmospheric column and the atmospheric structure is an important parameter to take into account for aerosol studies. Several measurement techniques are available for characterizing the aerosol properties and their vertical distribution in the atmospheric column. The Aerosol Robotic Network (AERONET, Holben et al., 1998), initiated by NASA's EOS, provides ground-based remote sensing aerosol measurements to follow the long-term aerosol properties such as optical, microphysical and radiative properties integrated over the whole atmospheric column at ambient conditions. Dubovik et al. (2000) performed sensitivity studies on the AERONET retrievals, showing difficulties to retrieve complex aerosol properties such as the single scattering albedo or the aerosol refractive index when the aerosol optical depth is low (AOD<0.2) but demonstrates that particle size distribution (PSD) are well retrieved for AOD higher than 0.05. For AOD ranging between 0.05 and 1.0, they calculate that an error of ±0.01 on direct AOD measurement at 440 nm can generate on error on Sun photometer volume concentration retrievals between 15 and 35% between 0.1 μm and 7 μm and between 30 and 100% out of this range.

*In-situ* measurements provide a large data set of aerosol properties: size distributions over a large size range, chemical composition, hygroscopic properties and optical properties. However, *in-situ* measurements performed from ground-based stations provide aerosol properties at a single point of the atmosphere, often under dry conditions. Boyouk et al. (2010) have studied the bias between ground-level aerosol mass concentrations and aerosol mass concentrations retrieved from both satellite and a Sun photometer during a 12 days period in Lille, France. Results show in-situ PM2.5 measurements 20% higher than the retrieved PM2.5 from Sun photometer and a correlation coefficient of 0.56 between the two measurement techniques. The authors highlight the fact that taking into account the inhomogeneity of the atmosphere can improve





significantly the correlation between ground level *in-situ* measurements and remote sensing retrievals. They show that correcting the Sun photometer retrievals by the ML height decreases the discrepancy between measurements by about 10%. Bergin et al. (2000) have also highlighted the importance of taking into account the ML height into the comparison of *in-situ* ground based (Southern Great Plains, Oklahoma, 320 m a.s.l) measurements (20 cloud free measurements) and remote

sensing extinction coefficients (from r²=0.55 to r²=0.78) using a one year period data set. The authors found integrated *in-situ* AOD 70% lower than Sun photometer AOD in dry condition and 40% lower taking into account Hygroscopic Growth Factor on extinction coefficient. The aerosol extinction coefficient distribution along the atmospheric column has also been investigated by Schmid works (Schmid et al., 2003; Schmid et al., 2006; Schmid et al., 2009) during several experiments at different locations (ACE-Asia in Japan, ARM AIOP (Ferrare et al., 2006) and ARM ALIVE in Oklahoma) with an airborne

Sun photometer (AATS-14), integrated airborne *in-situ* measurements and LIDARs (Raman LIDAR and Micro Pulse LIDARs). The experiments are based on the comparison of 19 flights during ACE-Asia program (Avril 2001), 16 flights during AIOP campaign (May 2003) and 12 flights during ALIVE campaign (September 2005). All studies report *in-situ* extinction coefficients lower than retrieved Sun photometer extinction coefficients (between 11% and 17% lower), and higher than LIDARs measurements (between 13% and 54% higher). Authors mainly explain the discrepancies between

measurements methods by humidity effects, by largest particles losses from *in-situ* probes and by the correction of aerosol vertical distribution in LIDAR profiles (mainly due to overlap correction).

A similar comparison performed during dust cases over Morocco concludes to similar discrepancies, with the *in-situ* (airborne measurements) AOD being around 20% lower than Sun photometer direct measurements at a wavelength of 500 nm from three different flights (Müller et al., 2012). However, the authors highlight significant uncertainties in aerosol

properties retrievals from Sun photometers which propagate to climate forcing modelling, especially when the atmosphere is inhomogeneous. Molero et al. (2012) have studied aerosol size distribution from ground based in-situ measurements and Sun photometer retrievals from a three weeks campaign, also using LIDAR measurements (SPALI10 in Spain in 2010). Results show a disagreement between volume concentrations measured from the two techniques but a well retrieved shape of the accumulation and coarse modes. Authors explain this discrepancy by the non-homogeneous atmosphere during the

comparison, in agreement with the conclusions from Müller et al., 2012. To our knowledge, comparisons between ground-based *in-situ* measurements and remote sensing retrievals have not yet been performed from high altitude sites over long term periods.

In the present paper, we compare the extinction coefficients and PSD retrievals from a low altitude Sun photometer with the same aerosol variables measured *in-situ* at a nearby high altitude station, using the information on the atmospheric structure

from a co-located LIDAR. We investigate the impact of the atmospheric structure and of the aerosol hygroscopicity on the agreement between the *in-situ* and remote sensing measurement techniques.



## 2. Site and instrumental description

The PUY atmospheric station is located in central France (45°77 N; 2°96 E) and composed of a high altitude site (at the top of the Puy de Dôme, 1465 m.a.s.l.) and a low altitude site (Cezeaux, 410 m a.s.l.). It is part of the global GAW (Global Atmospheric Watch) network, and one of the 43 ACTRIS (Aerosol, Cloud and Trace gases Research Infrastructure Network) stations for monitoring climate-relevant atmospheric variables. Meteorological parameters (temperature, relative humidity (RH), wind speed and direction) are continuously monitored at both sites (Puy de Dôme and Cezeaux). At the Puy de Dôme station (PUY), a large dataset of gaz-phase parameters ($O_3$, CO, $CO_2$, NO, $NO_2$, $CH_4$, $N_2O$, $SF_6$, VOCs, Rd), and of the particulate phase (scattering, absorption, Elemental and Organic Carbon (EC/OC) on filter, Inorganic ions on filter, Condensation Nuclei (CN)>10 nm, CN >1.2 nm, size distribution of nucleation mode (2-40 nm), size distribution of fine mode (10-400 nm), size distribution of coarse mode (0,5-10 microns), PM10, and Cloud Condensation Nuclei (CCN) size distribution) are continuously monitored. Previous work shows that the station is representative of the regional background (Asmi et al., 2011; Henne et al., 2010). The Puy de Dôme is located on the first mountain chain facing dominant western winds. It is one of the highest points of the Chaîne des Puys, which comprises 80 volcanoes aligned north to south on a 3 to 5 km wide strip of land, a little over 45 kilometres in length. This configuration induces less modifications of the general airflow than in larger mountain chains such as the Alps in Europe. Indeed, this feature leads to a quasi-absence of valley winds, observed in more complex topographies, and permits to sample air masses often representative of the altitude at which the station is located. 11 km East of and 1055 m below the Puy de Dôme, the Cézeaux University Campus site (CZ, 45°76 N, 3°11 E, 410 m a.s.l.) houses remote sensing measurements that give information on the structure and properties of aerosol layers along the atmospheric column. The PUY station is one of the very few high altitude stations worldwide measuring a complete set of *in-situ* measurements of the gas and particulate phases, coupled with nearly co-located LIDAR and Sun photometer measurements at its base.

### 2.1 *In-situ* aerosol measurements at the Puy de Dôme station

Aerosol particles are sampled through a temperature controlled Whole Air Inlet which higher size cut is 35 micron under average 6 m.s$^{-1}$ wind speed conditions. A gradient between ambient and room temperatures insures that the relative humidity monitored at the inlet of the instruments does not exceed 40% and that the aerosol is characterized at its dry state.

Aerosol Particle Size Distributions (PSD) are characterized with a combination of a Scanning Mobility Particle Sizer (SMPS, Venzac et al., 2009) and an Optical Particle Counter (OPC – Grimm 1.108 in 2011), covering the diameter range of 10nm-400nm and 350nm-18µm respectively with a total of 118 bins. The SMPS, developed at the Laboratoire de Météorologie Physique (LaMP) at Clermont-Ferrand, is composed of a neutralizing device, a Differential Mobility Analyzer (DMA) operating in a closed dried loop, and a Condensation Particle Counter (CPC). The SMPS is based on the scanning mobility measurement concept introduced by Wang and Flagan (1990). The OPC is operating with a laser beam at 638nm crossing a particle chamber, and retrieving the particle size distribution from the diffused light (Burkart et al., 2010).





Aerosol light absorption ($\sigma_{abs}$) and scattering ($\sigma_{scat}$) coefficients are measured using a Multi Angles Absorption photometer (MAAP 5012, 670 nm) and a three wavelengths (450, 550 and 700nm) nephelometer (TSI 3563) respectively. The MAAP instrument measures the radiation transmitted and scattered back from a particle-loaded fiber filter to retrieve an absorption coefficient (Petzold and Schönlinner, 2004). The nephelometer measures the integrated light scattered by particles from 7° to 170° and from 90° to 170°. Nephelometer data are corrected for detection limits and truncation errors according to Anderson and Ogren (1998). The angstrom coefficient (å) is computed from multi-wavelength nephelometer measurements according Eq. (1). This angstrom coefficient is used to compare all instruments at the same wavelength. In this study, the 675 nm channel corresponding to a Sun photometer wavelength is selected, 5 nm apart from MAAP measurements wavelength ($\lambda_0$=670 nm) and 25 nm from the nephelometer measurements wavelength ($\lambda_0$=700 nm). Hence, the in-situ extinction coefficient ($Ext_{IS}$) and aerosol single scattering albedo ($\omega_0$) are calculated at $\lambda$=675 nm according to Eq. (2) and Eq. (3).

$$\frac{\sigma_{scat}(\lambda)}{\sigma_{scat}(\lambda_0)} = \left(\frac{\lambda}{\lambda_0}\right)^{-\text{å}} \tag{1}$$

$$Ext_{IS}(\lambda) = \sigma_{abs}(\lambda) + \sigma_{scat}(\lambda) \tag{2}$$

$$\omega_0(\lambda) = \frac{\sigma_{scat}(\lambda)}{\sigma_{scat}(\lambda) + \sigma_{abs}(\lambda)} \tag{3}$$

### 2.2 Remote sensing measurements at Cézeaux site.

A CIMEL Sun photometer (CE-318), operating at the CZ site, measures the aerosol optical properties of the total integrated atmospheric column under ambient conditions at four wavelengths (440, 675, 870 and 1020 nm). The instrument is part of the AErosol Robotic NETwork (AERONET; Holben et al., 1998). Data are automatically sent to the Laboratoire d'Optique Atmsopherique (LOA at Lille) for processing and full inversion data are available on the AERONET web site. Direct Sun photometer measurements provide the column averaged Aerosol Optical Depth (AOD) and angstrom exponent. Additionally, an inversion algorithm described by Dubovik and King, (2000) is used to retrieve the particle volume size distribution on the 0.10 to 30µm diameter range. The version 2 and level 1.5 of AERONET data is used in this study corresponding to automatically cloud filtered data. The Aerosol PSD retrieved from the Sun photometer measurements is meanly separated into two modes, a fine mode (particles <0.6µm) and a coarse mode (particles >0.6µm) with a possible intermediate mode (Kaufman and Holben, 1996). A discussion on errors of the data products can be found in Dubovik et al. (2000).

A LIDAR provides information on the vertical profile of aerosol particle properties continuously. The LIDAR operated at CZ is a Raymetrics system with a laser emitting a polarized light at 355 nm and a telescope of 40cm diameter collecting the light backscattered by molecules and aerosols. An optical box allows to separate the Rayleigh-Mie signal at 355 nm in two perpendicular polarized directions and the Raman signals due to nitrogen (387 nm) and water vapor (408 nm). The minimum time resolution of one profile is 1 min and the vertical resolution is 7.5m. The measurements follow the procedures of the European Aerosol Research Lidar Network (EARLINET). For the present study, a Klett inversion (Klett, 1985) is used to retrieve elastic aerosol backscatter profiles with a constant LIDAR ratio of 58sr (Müller et al., 2007), corresponding to a



mean Central Europe air mass LIDAR ratio at 355 nm. A more detailed description of this instrument and of the data inversion are available in Hervo et al., (2012) and Freville et al. (2015). In this study, the LIDAR profiles are mainly used to detect multiple aerosol layers and to evaluate the optical contribution of aerosols in the different atmospheric layers (ML and FT). In order to correct the blind zone between the laser beam and the field of view of the telescope, a theoretical overlap
(Kuze et al., 1998) is applied on LIDAR measurements from the ground to the altitude of full overlap. Tests on this theoretical overlap were performed by comparing the altitude of the range corrected LIDAR signal maximum ($Pr^2_{max}$) with the theoretical altitude of full overlap retrieved using the telecover technique. The comparison between $Pr^2_{max}$ and the full overlap altitude gives a correlation coefficient ($r^2$) of 0,997 and with a slope close to 1. This permits to make a good estimation of the full overlap altitude only by the knowledge of the $Pr^2_{max}$.

### 3.  Data analysis

In the present study, the aerosol Particle Size Distributions (PSD) obtained from *in-situ* measurements and columnar remote sensing measurements are compared. The SMPS PSD can be fitted with a sum of log-normal modes to retrieve Nucleation, Aitken and Accumulation modes. However, for the purpose of comparison to the size distribution retrieved from the Sun
photometer, we chose to fit the SMPS size distribution with a single submicron mode, while the supermicron mode is fitted on the measured OPC PSD (Figure 1). The Sun photometer PSD is converted from $\mu m^3.cm^{-2}$ to $\mu m^3.cm^3$ dividing concentrations by the Mixing Layer height from LIDAR measurements. The two modes from the two different instruments are considered separately in order to increase the number of cases available for comparison. The dry PSD is then corrected to ambient humidity using a Hygroscopic Growth Factor (HGF) parametrization.

Hervo et al. (2014) have shown at the PUY station a strong impact of hygroscopicity on aerosol optical properties depending on air mass origin and season. In this study, in order to take into account the humidity effect on aerosol properties, the HGF is applied to dry *in-situ* diameters in order to retrieve wet concentrations for each mode. A seasonally segregated parameterization (Holmgren et al., 2014) initialized at PUY station, is used to apply an average HGF to the measured *in-situ* dry PSD. As the Puy de Dôme and Cézeaux stations are separated by 11km and 1km of altitude difference, a mean relative
humidity between the two sites is used in the parametrization. Because the parametrization is not well adapted for particles smaller than 30nm and higher than 420nm, a HGF of 1 is taken for small particles (<30nm) and the HGF at 420nm is applied to higher particles (Rose et al., 2013). The wet distribution is then fitted to retrieve wet concentrations and mean diameters for the two modes.

The comparison of the *in-situ* and remote sensing aerosol properties is strongly dependent on the atmospheric structure and
layering. Hence, the STRAT (Structure of the Atmosphere) algorithm (Morille et al., 2007) is applied to the LIDAR measurements to identify atmospheric structure as cloud, molecular and aerosol layers, and also to give information on noise



detection. The ML height is retrieved using the Wavelet Covariance Technique (WCT, Brooks, 2003; Baars et al., 2008; Hervo et al., 2014) on the range corrected LIDAR signal.

Because the Sun photometer measures integrated aerosol properties and the PUY station is either in FT conditions or in ML conditions, it is necessary to separate optical contributions of ML and FT from the Sun photometer signal. Using the ML

height obtained from the LIDAR backscatter profile (β), the fraction of aerosols from the ML that contributes to the total aerosol signal (over the whole atmospheric column) is calculated. In the present study, ML contribution ($R_{ML}$) is calculated from LIDAR backscatter (β) according Eq. (4) and permits to separate both ML and FT properties from the Sun photometer signals according Eq. (5) and Eq. (6).

$$R_{ML} = \frac{\beta_{ML}}{\beta_{column}} \tag{4}$$

$$AOD_{ML} = AOD_{TOTAL} * R_{ML} \tag{5}$$

$$AOD_{FT} = AOD_{TOTAL} * (1 - R_{ML}) \tag{6}$$

Where $\beta_{ML}$ is the LIDAR backscatter coefficient integrated from the ground level to the ML height. $B_{column}$ is the LIDAR Backscatter coefficient integrated from the ground to an upper altitude of 5km, above which it is assumed that the aerosol contribution to the total backscatter is negligible (Nicolas et al., in prep). This fraction is then applied to the total AOD

measured by the Sun photometer ($AOD_{TOTAL}$) to calculate the AOD of the ML ($AOD_{ML}$) (eq. 5) and the AOD of the FT ($AOD_{FT}$) (eq. 6).

Figure 2 shows an example of aerosol backscatter time series from LIDAR measurements, ML Height and $R_{ML}$ the 29[th] of September 2011. On the top, green and black points show possible LIDAR signal inversion and missing data respectively. The dotted black line on both panels represents the altitude of the PUY station. For this case, ML Height varies from 1200

m.a.s.l. during nighttime to 1500 m.a.s.l. during daytime corresponding to a small heat convection (autumn case). On this day, the ML contribution shows a diurnal variation with a maximum during daytime, showing that aerosol sources increase during daytime and are not fully compensated by dilution when the ML extends. For this typical case, the ML contribution varies from 60% at night to 78% shortly after noon (UTC) in agreement with previous results. Ricchiazzi et al. (2006) found ML contributions between 45% and 90% using airborne photometer (AATS-14) profiles during 4 free cloud days and ML

contributions between 19% and 72% (mean of 44%) are found by Bergin et al. (2000) using a LIDAR profile technique.

## 4. Dataset used

Long-term data sets present the advantage of offering a large variability of statistically reliable environmental cases, and allow to investigate contrasts between seasons, time of the day, meteorological conditions, or air mass types. In this study, we focus on the year 2011 that offers a large availability of simultaneous measurements. Figure 3 shows the availability time



series of each instrument needed for this study (LIDAR, Sun photometer, SMPS or OPC, MAAP, Nephelometer and meteorological parameters) over the 2011 period. *In-situ* Relative Humidity (RH) above 95% is used as a cloud screening, in addition to the cloud classification derived from the STRAT algorithm applied to the LIDAR measurements. Despite the need of a large amount of instruments operating simultaneously under clear sky conditions, 357 1-hour data points can be

used for extinction comparisons and 116 1-hour data points if multilayers cases are excluded using STRAT. For particle size distribution comparisons, 412 1-hour data points are available and 155 data points (black in Figure 3) if multilayers cases are excluded using STRAT. Among these 155 data points, 92 include SMPS size distributions and 80 include OPC-GRIMM size distribution.

No OPC-GRIMM measurements have been acquired on January and from August to September, and an important gap is

observed in August and September due to Sun photometer issues. Despite this lack of data, all season can be represented in this study with a significant number of cases as shown on Figure 4.

## 5.  Results

### 5.1  AOD and aerosol extinction

#### 5.1.1  Cloud-free conditions

A first comparison of direct measurements of the aerosol *in-situ* extinction and remote sensing (Sun photometer) AOD can be performed. For this comparison, we selected cloud-free measurements which represent 357 data points.

Figure 4a) shows the relationship between the aerosol extinction measured from *in-situ* probes at dry state and the AOD measured by the Sun photometer according the season at 675 nm. A closer view on the relationship at low extinctions is shown **Error! Reference source not found.**. Only few points during summer 2011 are taken into account due to

nstrumental issues (43 points during summer, 80 during autumn, 86 during winter and 148 during spring). Although the AOD is integrated over the whole atmospheric column while the aerosol in-situ extinction is measured at one single altitude, they are strongly correlated ($r^2$=0.82), indicating that the intermediate altitude of 1465 m, often at the interface between the ML and the FT where *in-situ* measurements are performed is overall representative of the whole atmospheric column. This results shows a better correlation between measurement techniques than the correlation reported in previous works ($r^2$=0.55

between Sun photometer AOD and ground base in-situ extinction coefficient according Bergin et al. (2000)). This result might be explained by a better representativeness of the atmospheric column by high altitude *in-situ* measurements. No clear difference is observed between seasons. In order to quantitatively compare aerosol extinctions measured by *in-situ* and remote sensing techniques, a first approximation is to assume that most of the Sun photometer signal is due to aerosols present in the ML. Hence, we calculate the average aerosol extinction as the AOD contained in the ML following:

$$Ext_{Sun\ Photometer} = \frac{AOD_{Sun\ Photometer}}{ML\ Height} \tag{7}$$



Where ML height is obtained from the LIDAR data.

Figure 4b) shows the aerosol extinction coefficient measured from *in-situ* instruments as a function of the Sun photometer aerosol extinction coefficient derived from *eq. (7)*. The *in-situ* aerosol extinction coefficients are still highly correlated with the Sun photometer extinction coefficient ($r^2$=0.73). This results is in a good agreement results reported by Bergin et al.

(2000), showing a correlation factor of 0.78 between *in-situ* and Sun-photometer measurements. A lower correlation than with the total AOD can be attributed to the fact that on one hand, the Puy de Dôme instruments are not always sampling the ML but also frequently the FT as shown on Figure 2 (bottom), and on another hand, not all of the AOD is due to the ML. The *in-situ* extinction is on average only 14 % of the Sun photometer Extinction (12% during winter, 17% during summer, 22% during autumn and 24% during spring). This is first likely due to the fact that a number of *in-situ* data are especially

low because they are measured in FT conditions. Indeed, characterized by the lowest ML Height, winter cases present the largest difference between the two measurement techniques. Moreover, other bias between the two measurements may be due to the fact that on some occasions such as during dust episodes, high concentrations of aerosols are preferentially transported at high altitudes, as evidenced in Bourcier et al. (2011). In some cases, dust or sea salt aerosols may be transported above the Puy de Dôme station, and hence captured by the Sun Photometer but not detected by the Puy de Dôme

*in-situ* instrumentation.

### 5.1.2    Impact of a multi-layer atmosphere

In order to exclude such cases of multi-layer aerosol transport, the STRAT algorithm was applied to filter them out from the database.

After filtering the multilayer cases, only 116 cases (33%) remain from the original data set. Due to this lower number of data

points, the correlation between the *in-situ* and Sun photometer extinction is lower when multilayer cases are excluded ($r^2$=0.59) (Figure 5). Under these particular atmospheric conditions, the *in-situ* extinction coefficient at the PUY station is ranging from 0 to 0.04 km$^{-1}$ while the Sun photometer extinction ranges from 0 to 0.2 km$^{-1}$. The Sun photometer values higher than 0.2 km$^{-1}$ observed on Figure 4 but not on Figure 5 were indeed due to heterogeneous cases, thus probably corresponding to dust or marine aerosol transported at higher altitudes than the Puy de Dôme station. As these points

corresponding to multilayer cases were still well correlated, it is likely that the Puy de Dôme *in-situ* measurements capture high altitude aerosol transport events, but at a diluted concentration. After multilayer filtering, the *in-situ* extinction is closer to the extinction from the Sun photometer (25% of the Sun photometer extinction compared to 14% before multilayer filtering).

The second likely explanation for measuring a lower extinction from *in-situ* instruments compared to remote sensing average

is the fact that the Puy de Dôme station is sampling in a less concentrated aerosol layer (FT) a part of the times. This is evidenced when color-coding *in-situ* extinction measurements by their belonging to the ML or to the FT (Figure 5). A threshold at 1200 m a.s.l. is taken for ML conditions at PUY station in order to take into consideration the impact of forced





convection due to topographic effects (Venzac et al., 2009). The results clearly show different correlations between extinction coefficients when *in-situ* measurements are performed in the ML (in red, WCT > 1200m) and when they are performed in the FT (in blue, WCT < 1200m). In FT conditions, *in-situ* measurements are mainly lower than the overall fitted line and, in ML conditions, *in-situ* measurements are mainly higher (33% of the Sun Photometer measurements in ML

conditions and 20% in FT conditions).

### 5.1.3    Impact of the layer contributions

When the PUY site is in the ML, the fraction of the Sun photometer AOD comprised in the ML (Eq. (5)) was calculated by using the ratio of ML/Total backscatter measured by LIDAR following Eq. (4) (Sect. 3). Using the same method, when the Puy de Dôme *in-situ* measurements are performed in the FT, the corresponding fraction of the Sun photometer AOD is

calculated following Eq. (6).

Results are shown on Figure 6. A global analysis combining all data (ML and FT cases together) shows a net improvement of the agreement between measurements, with the Sun photometer extinction being 49% lower than the *in-situ* extinction. This discrepancy is still higher than the one previously reported in the literature between extinction coefficients measured by a Sun photometer and *in-situ* airborne probes, performed on case studies. Several studies found in-situ measurements being

around 15% lower than remote sensing measurements (Schmid et al., 2003; Schmid et al., 2009; Müller et al., 2012). Our results are not issued from *in-situ* complete vertical profiles, but from one single point measurements, which likely explains the different slopes and the higher discrepancy. In details, the 116 data points were separated between cases when the PUY station was located in the ML (63 data points, red fitted curve and circle markers) and when it was located in the FT (53 data points, blue fitted curve and cross markers) based on the WCT calculation from LIDAR measurements (Figure 5). For ML

cases, the Sun photometer measurements are closer to *in-situ* extinction than in the previous analysis (*in-situ* extinction values represent 45% of the ML Sun photometer extinction as opposed to 33%). For FT cases, the *in-situ* extinction is more than 2 times higher than the FT Sun photometer extinction. These results are mainly explained by the presence of aerosol concentration gradients in both atmospheric layers, which are not fully well mixed. *In-situ* probes are measuring either in the upper part of the ML, where concentrations are not as high as at surface where sources of aerosols are located, or in the

lower part of the FT, more influenced by the ML concentration than the upper FT. To summarize, when PUY station measurements are in ML conditions, optical properties would be underestimated compared to the mean ML and vice-versa in FT conditions.

The comparison of extinctions in Figure 6 is coloured by the mean relative humidity (RH) between the two stations which is a good tracer for ML height (Seidel et al., 2010). The goal was to examine if an additional explanation for the *in-situ*

measurement to be lower than the Sun photometer data in the ML can be the amount of condensed water contributing to the aerosol extinction, which is not taken into account in the *in-situ* measurements. We observe that even though FT air masses (cross marker on Figure 6) are indeed overall dryer than ML air masses, in each atmospheric layer the high relative humidity



data point do not obviously correspond to dispersed data points. Although previous study have shown an important role of humidity in the comparison of *in-situ* and remote sensing measurements (in reducing the discrepancy from 70% to 40%, Bergin et al., 2000), the water content of aerosol do not seem to be the main factor influencing the discrepancy between *in-situ* and remote sensing extinctions in the present study. However, we will estimate this effect in a more quantitative approach when comparing *in-situ* and Sun photometer retrievals of the aerosol PSD.

## 5.2 Particle Size Distribution

### 5.2.1 Atmospheric structure filters application

The *in-situ* volume PSD (SMPS and OPC) were compared to the volume PSD retrieved from the Sun photometer measurements for both coarse and fine modes. The comparison with the totality of the Sun photometer and *in-situ* PSD parameters regardless of the situation of the PUY station in the FT or ML can be found in the supplementary (Figure A.9). The *in-situ* fine mode diameters are clearly higher (by 72 % for the majority of cases) than the diameters retrieved with the Sun photometer for all seasons while the *in-situ* concentrations are mainly lower than the concentrations retrieved from the Sun photometer, in agreement with the extinction measurement comparison (Figure 4). As evidenced by the comparisons of extinction coefficients, the PSD parameter comparison can also be biased by the fact that the *in-situ* data is for some periods representative of the FT only, while the Sun photometer inversion takes the whole column into account. Figure 7 shows results of the comparison between *in-situ* and Sun photometer inversion parameters only for homogeneous cases, when *in-situ* data points are segregated into the ML and FT, and the corresponding layer heights are taken into account: concentrations retrieved from the Sun photometer were calculated per unit volume of boundary layer for ML cases (divided by the MLH) and of free tropospheric layer for FT cases (divided by 5 000 m - MLH). 73 data points are available for fine mode concentrations and 66 points for the coarse mode concentrations. As observed for the extinction, taking into account the segregation between ML and FT and their respective contributions is a main factor influencing the discrepancy between the two measurements techniques.

The average ratio between diameters measured by both techniques is in the range 0.77-1.45. Overall, *in-situ* diameters in the fine mode are 44% higher than the diameters retrieved from the Sun photometer. Previous detailed analysis of the *in-situ* submicron aerosol size distribution show that this mode is characterized by two modes, Aitken (diameters between approximately 10 and 100 nm) and Accumulation (diameters between approximatively 100 and 1000 nm) modes (Venzac et al., 2009) that are merged into one single mode during the fitting procedure. Since aerosol scattering is more sensitive to the accumulation mode than to the Aitken mode (Figure A.10), Sun photometer retrievals might be biased towards this accumulation mode diameter. As consequence, Sun photometer fine diameters should be overestimated due to the important influence of the Accumulation mode compared to the Aitken mode. Because in-situ fine mode diameters are already higher than Sun photometer fine mode diameters, the bias between the two measurement techniques may even be more important





than observed. Concerning the coarse mode, *in-situ* diameters are in relatively good agreement with the Sun photometer retrievals (slope of 1.07 in the ML and 0.77 in the FT). Coarse mode diameters are better correlated and in better agreement when the PUY station is in the ML than when it is in the FT.

In the ML cases, the *in-situ* aerosol volume concentrations are 80% and 54% of the Sun-photometer retrievals for fine and
coarse modes respectively (as opposed to *in-situ* extinction being 45% of the Sun photometer extinction). Fine mode volume concentration comparisons are in good agreement with Schmid et al. (2003) reporting airborne *in-situ* extinction between 0 and 4 km a.s.l. 13% lower than Sun photometer extinction. However, the correlation coefficient in ML conditions is significantly lower ($r^2<0.60$) for volume concentrations compared to the extinction coefficient analysis ($r^2=0.79$).

In the FT cases, *in-situ* extinction coefficient was 2.26 times higher than Sun photometer extinction, while it is 22% higher
and 32% lower for fine and coarse mode aerosol volume concentrations respectively. Contrary to the ML cases, the correlation observed in the FT cases between volume concentrations increased in comparison to extinction coefficients study ($r^2=0.43$ and 0.61 for fine and coarse modes respectively as opposed to 0.30 for extinctions comparison). This would indicate that the inversion procedure for retrieving the PSD from Sun photometer measurements is relatively reliable when the ML height is low and the Sun photometer measurement mainly represents the upper atmospheric variations (FT cases), but less
reliable when the ML is well developed. The fact that the slopes of *in-situ* to Sun photometer aerosol volume concentrations are lower than the slopes of the *in-situ* to Sun photometer extinction for FT cases indicates a slight overestimation of the aerosol volume concentrations (for both fine and coarse modes) due to the Sun photometer retrieval procedure when the ML height is low.  On the contrary, the Sun photometer retrieval would slightly underestimate the aerosol volume concentrations (for both fine and coarse modes) when the ML is well developed (ML cases).

20          ### 5.2.2    Hygroscopicity impact

In the present section, we will quantify the impact of hygroscopicity on *in-situ* vs remote sensing comparison. Figure 8 shows volume PSD parameters after *in-situ* PSD have been corrected for their hygroscopic growth at ambient relative humidities as described in Sect. 3 for both ML and FT cases. The application of the HGF on volume PSD significantly influences the fine mode PSD parameters and a clear increase of diameters and concentrations are observed for high RH
cases (RH>50%). The bias between *in-situ* and remote sensing diameters are about 40% higher than for the dry cases in the ML and in the FT. For fine mode volume concentrations, the bias between *in-situ* and remote sensing is more than 70% higher than in dry cases in the FT and *in-situ* volume concentrations became higher (by 20%) than the Sun photometer volume concentrations in the ML. Because the same value of HGF is applied for particles above 420nm (see Sect. 3), the effect of humidity on coarse mode PSD parameters is more uncertain. Aerosols may be more hygroscopic than predicted by
the parametrization if they are sea salt aerosols, or less hygroscopic than predicted if they are Saharan dust. However, one can observe that applying a hygroscopic growth factor to the coarse mode *in-situ* PSD brings the comparison to 20% larger



*in-situ* than Sun photometer diameters for the ML cases, and 14% lower *in-situ* than Sun photometer diameters for the FT cases. No significant change is observed for coarse mode volume concentrations. Coarse mode in situ concentrations are in better agreement with the retrieved coarse mode concentrations when the hygroscopic growth is taken into account as well. Moreover, at PUY station, most of coarse mode particles observed are Saharan dusts that are barely hygroscopic.

## 5 Conclusion

The continuous measurements of aerosol properties at the Puy de Dôme station, 1465 m a.s.l., in parallel to remote sensing measurements performed at the Cezeaux site, 420 m a.s.l., allow to analyse the differences between the two measurement techniques. A one year data set of 6 different instruments operating simultaneously in synergy is used. To our knowledge, it is the first time that such a comparison is realized on a long term period. The comparison of *in-situ* and Sun photometer extinction measurements showed that the PUY station is representative of the total atmospheric column. For all seasons and for all extinction ranges, the two measurement techniques are well correlated. However, an important bias (86%) between *in-situ* and Sun photometer measurements is observed when the entire Sun photometer signal is hypothesised to be confined in the mixed layer (ML) and the multilayer cases are not filtered, especially when the ML height is low.

The bias is lowered when taking into account the heterogeneity of the atmosphere and the vertical atmospheric structure to separate ML and FT conditions. Using a combination of LIDAR (STRAT algorithm) and *in-situ* measurements, the one year data set was filtered from clouds and high altitude thin plumes from different sources. This first filter reduces significantly the number of data points (33% of initial data set for extinction and 30% for size distributions) and decreases the correlation factor. However, filtering multiplayer cases and correcting the Sun photometer extinction for either the ML or FT contribution reduces significantly the bias between *in-situ* and Sun photometer measurements (from 86% to 55% for ML cases).

At dry state, the particle size distribution comparison show fine mode diameters 44% higher from *in-situ* measurements than from Sun photometer retrievals and fine mode concentrations 20% lower and higher for ML and FT cases respectively in agreement with previous studies. In comparison to extinction bias, results permit to highlight the overestimation of aerosol volume concentrations in FT retrieved by remote sensing techniques for both fine and coarse modes.

The study also focuses on the impact of the humidity on size distributions parameters. A Hygroscopic Growth Factor is calculated from a seasonal parameterization in function to the relative humidity and applied to the in-situ particle diameters. In particular, results show an important impact of the humidity on the fine mode of the distribution which increases the bias between the two measurement techniques for diameters. The impact of aerosol hygroscocity on the extinction coefficient would be an interesting result using a Mie calculation but will need more information on the particle composition.

This long term study has shown that a ground-based site with *in-situ* aerosol measurements can be representative of the overall atmospheric column and of the regional conditions. Hence, the important number of instruments installed at the



mountain station gives the chance to characterize in a complete view different aerosol layers, taking into account that the extensive aerosol variables (concentration, extinction, scattering and absorption coefficients) are likely underestimated compared to the whole ML and overestimated compared to the whole FT. On the other hand, this study also validates complexe remote sensing retrievals such as the PSD, provided that the atmospheric aerosol structure is characterized and aerosol loading well distributed into the ML and FT. In particular, we show that the hypothesis of the totality of the aerosol loading comprised within the ML would lead to important errors if used in modelling exercises.

**Acknowledgments:** The authors would like to acknowledge the OPGC and its staff and INSU-CNRS for their contribution to establishing and maintaining the PUY measurement site. This work was performed with the financial support of the French ORAURE SOERE, the French national program SNO-CLAP, the European Infrastructure Projects ACTRIS (Grant N°262254) and ACTRIS-2 (Grant N°654109), and the support of the CNES EECLAT project.





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





**Figures :**

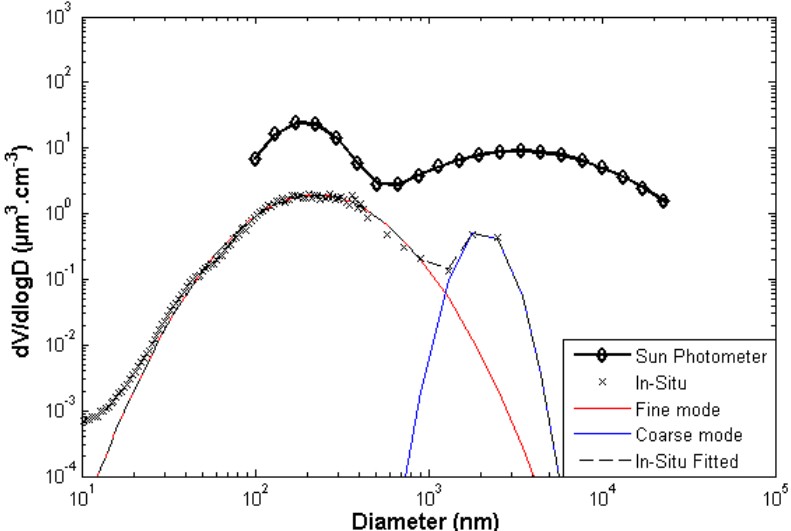

**Figure 1: Volume Particle Size Distribution from both *in-situ* (cross) and Sun photometer (bold line) measurements for the 7$^{th}$ of February 2011 at 11:03. *In-situ* measurements (SMPS and OPC) are fit according 2 modes, one fine mode (red line) and one coarse mode (blue line). The dashed black line is the sum of the two in-situ modes.**




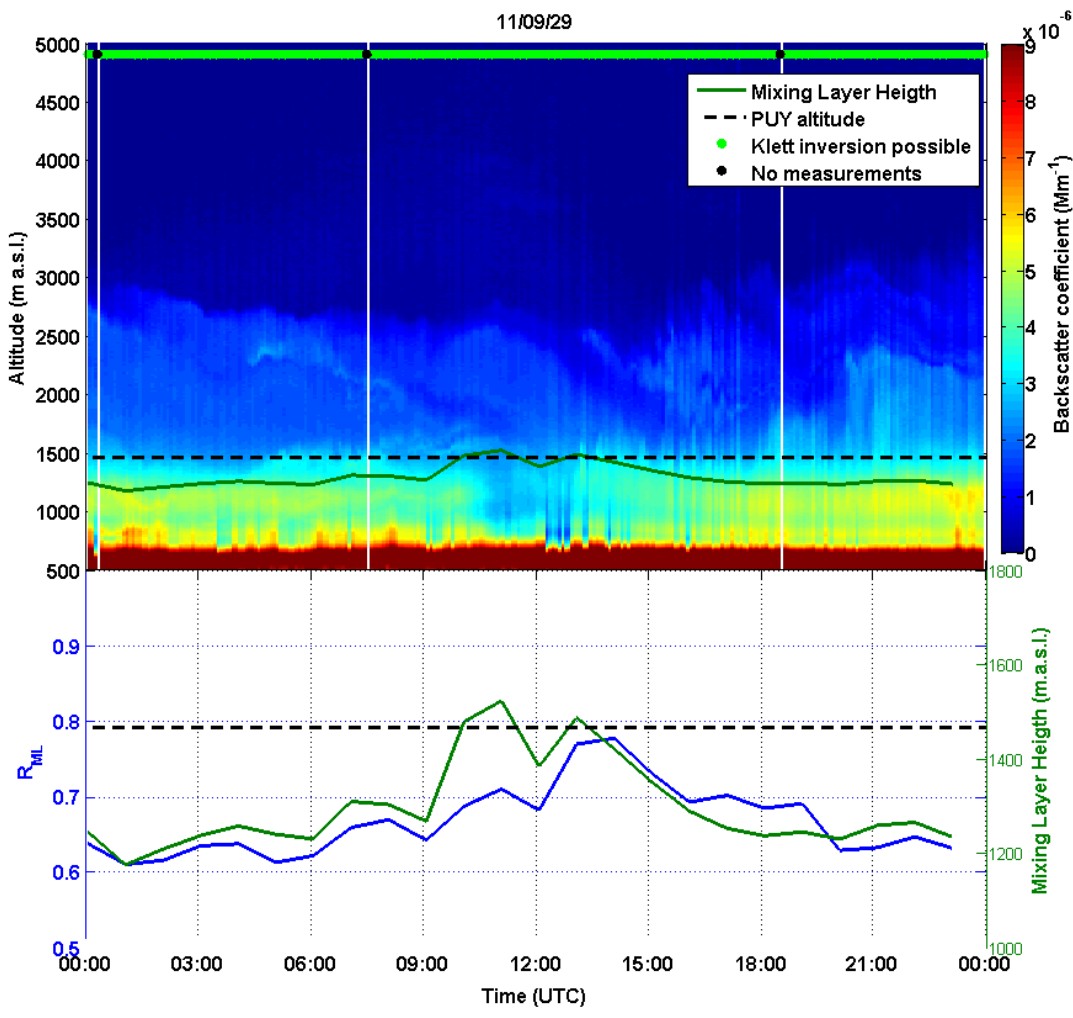

**Figure 2: Top: Aerosol backscatter time series (UTC) on 29 September 2011 from LIDAR measurements performed at CZ with a 5 min resolution. Green points at the top of the figure indicate measurements with possible inversion and black points indicate no LIDAR measurements. Black dashed lines represent the Puy de Dôme top. Bottom: Mixing Layer Height (green line) from WCT algorithm and Mixing Layer contribution ($R_{ML}$) (blue line).**





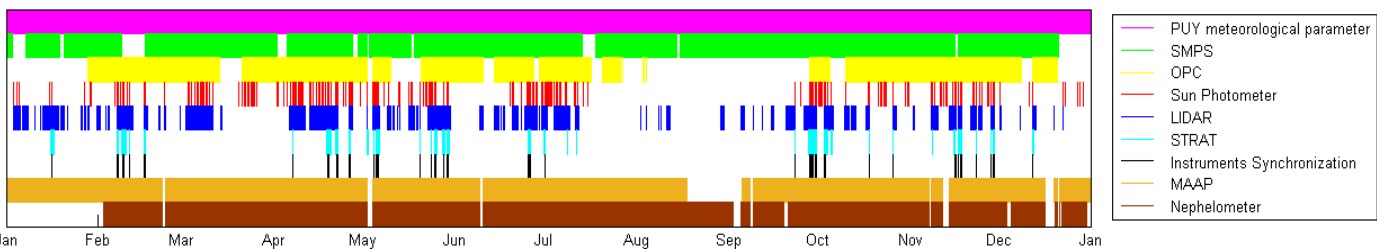

**Figure 3 : Temporal series of available instruments and cloud screening from RH information during the year 2011. Magenta, green, yellow, red, blue, orange and brown areas representing meteorological parameters at PUY station, SMPS, OPC-GRIMM, Sun photometer, LIDAR, MAAP and Nephelometer date. Cyan area show STRAT selection from LIDAR analysis and black area give selected points for this study.**

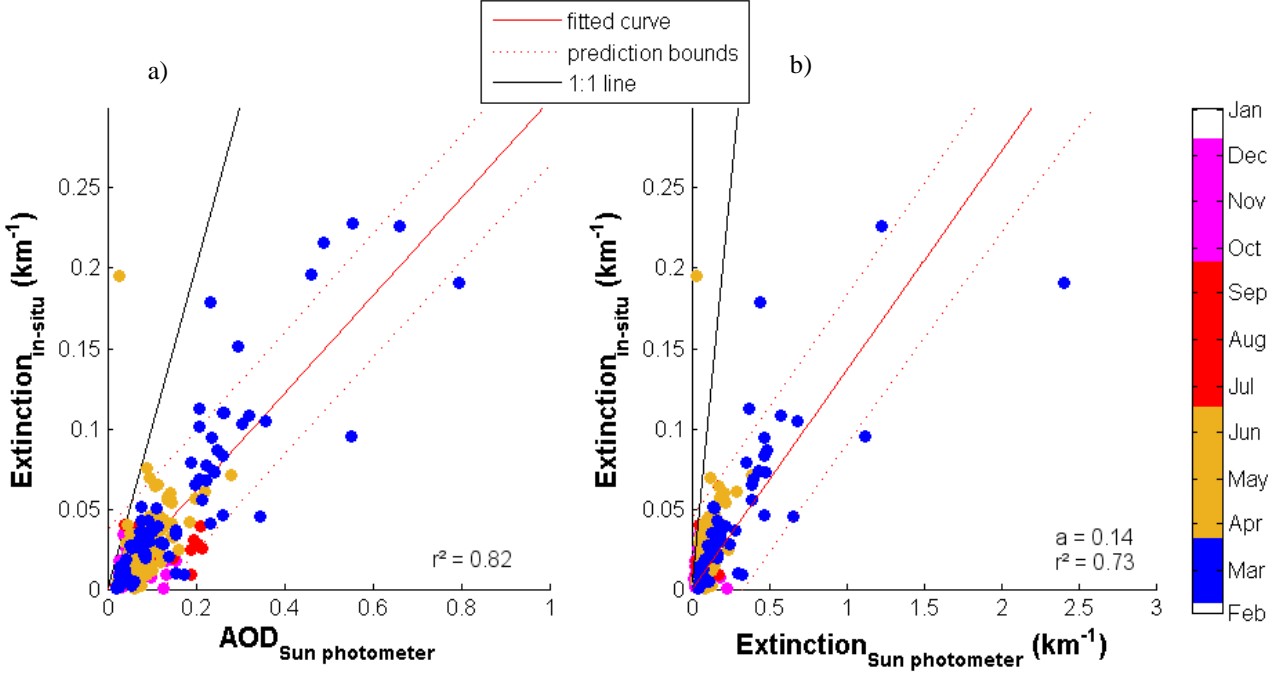

**Figure 4 : In-situ extinction coefficient at 675 nm calculated from nephelometer and MAAP measurements versus a) Sun photometer AOD and b) Sun photometer extinction (km$^{-1}$) color-coded by season. Full and dashed red lines correspond to the fitted line and prediction bounds respectively and the black line to the 1 to 1 line.**





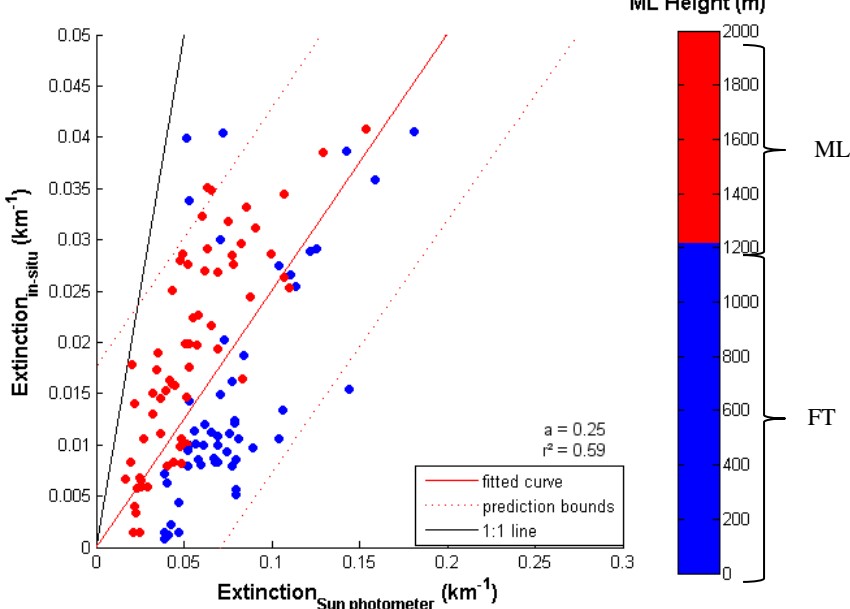

**Figure 5: Scatter plot of in-situ and Sun photometer extinction coefficients at 675 nm after filtering the multilayer cases of high altitude transport using STRAT algorithm colored by the ML Height (from WCT algorithm).**

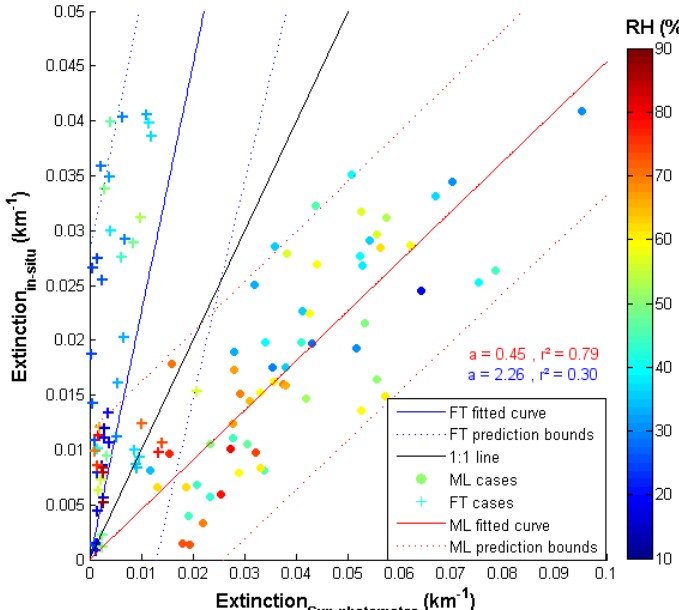

**Figure 6: Scatter plot of in-situ and Sun photometer extinction coefficients at 675 nm after applying the Mixing Layer contribution factor to Sun photometer measurements. Colors correspond to the mean Relative Humidity between the two sites. Blue fit and cross markers are for FT data and red fit and circle markers for ML data.**





**Figure 7: Scatter plot of volume PSD parameters after applying the ML and FT contribution factors to Sun Photometer concentrations. Blue fit and markers are for FT data and red fit and markers for ML data.**



**Figure 8: Scatter plot of volume PSD parameters after applying the Humidity growth factor (HGF). Color corresponds to the mean relative humidity between the two stations. Blue fit and cross markers are for FT data and red fit and circle markers for ML data.**





**Appendix A:**



**Figure A.9: Scatter plots of volume PSD parameters from in-situ and Sun photometer measurements for both fine (top panels) and coarse (bottom panels) modes in function of season. Left panels represent particle diameters comparison and right panel concentrations. Full red line represents the linear fit and dotted red lines the 5th and 95th percentiles. The black line is the 1:1 line.**




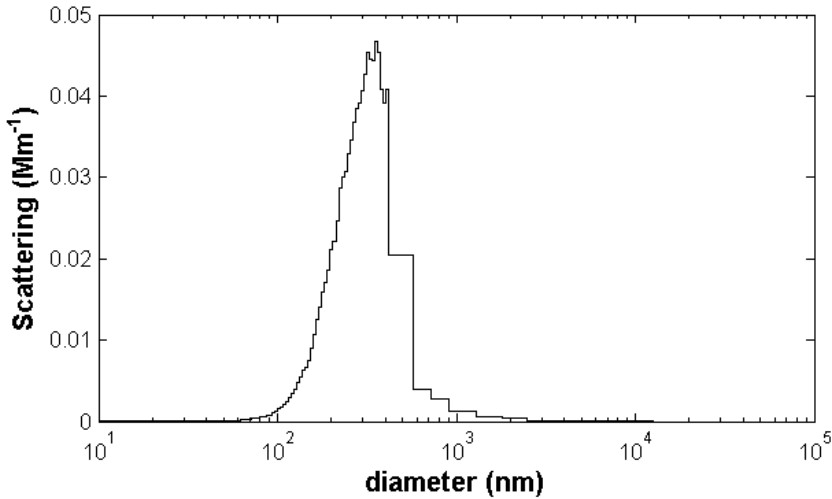

**Figure A.10: Scattering coefficient from Mie calculation according each bin of the Particule Size Distribution.**