# Peer review of "Comparison of the aerosol optical properties and size distribution retrieved by Sun photometer with in-situ measurements at mid-latitude."

_Atmospheric Measurement Techniques, 2016_

## Referee Comment (RC1) · Anonymous Referee #2 · 3 Jun 2016

A set of papers presents analysis of correlation between parameters of aerosol particles received from in-situ measurements at the ground-based sites or at meteorological towers and the results of columnar or altitude- resolved remote sensing data. Nevertheless, this multiple-factor problem remains relevant for interpreting data of complex atmospheric experiments. The manuscript by Chauvigné et al presents novel results for characterization of this problem because of uniqueness of the database gathered at the PUY atmospheric station. The authors analyze the results of long-term in-situ meteorological, aerosol optical and microphysical observations at two sites with altitude 0.4 and 1.4 km, as well as data of lidar and sun-radiometer measurements. Depending on meteorological conditions, the upper site of in-situ measurements was located either within mixing layer, or in free troposphere which allows the different variants of aerosol investigations to be considered. Paper can be recommended for publication.

List of specific comments:

Page 8, Line 18, 19: Remove "Error! Reference source not found" and correct the text. Page 8, Line 26: It is appropriate to note that results of in-situ measurements in the ground-based layer are affected by local aerosol sources. Measurement conditions at the high altitude Puy de Dôme site eliminate such interferences.

Figure 7 and 8: what means Diameter? (mode, median ....of fine/coarse particle volume distribution ?)

Part 5.2 The significant difference between "in-situ and Sun-photometer diameters" of fine particles is a very interesting result. Do authors have a physical interpretation of this feature?

---

## Referee Comment (RC2) · Anonymous Referee #3 · 4 Aug 2016

This manuscript refers to studies analyzing the results of comprahansive aerosol experiments. The authors present the data of long-term in-situ aerosol optical and microphysical observations at PUY atmospheric station (1465 m a.s.l.), as well as data of lidar and sun-radiometer measurements at Cezeaux University Campus site (410 m a.s.l).

List of comments is given below.

1. The paper reports the results of the comparison of the aerosol optical and microstructural characteristics in the atmospheric column and at altitude where the

PUY site is located. At PUY site aerosol light absorption ($\sigma$abs) and scattering ($\sigma$scat) coefficients are measured using a Multi Angles Absorption photometer and a three wavelengths nephelometer respectively. Similar equipment is installed on the aircraft to provide vertical profiles $\sigma$abs(z), $\sigma$scat(z) and extinction coefficient $\sigma$ext(z)=$\sigma$abs(z)+$\sigma$scat(z). The analysis of measurements of aerosol characteristics in the atmospheric column and their vertical profiles obtained on board of aircraft is presented in a number of studies (see **). Several papers also discusses the effect of relative humidity on the absorption $\sigma$abs(z) and scattering $\sigma$scat(z) coefficients. This manuscript does not consider the vertical profiles of the extinction coefficient, however, some approaches and results (** and others) logical to use in current study. Is possible it will be useful in the interpretation of data and help to explain the significant difference between in-situ and Sun-photometer measurements. The above also applies to the particle size distribution function.

**

Elias, T., S. J. Piketh, R. Burger, and A. M. Silva, Exploring the potential of combining column-integrated atmospheric polarization with airborne in situ size distribution measurements for the retrieval of an aerosol model: A case study of a biomass burning plume during SAFARI 2000, J. Geophys. Res., 108(D13), 8508, doi:10.1029/2002JD002426, 2003.

Haywood, J., P. Francis, O. Dubovik, M. Glew, and B. Holben, Comparison of aerosol size distributions, radiative properties, and optical depths determined by aircraft observations and Sun photometers during SAFARI 2000, J. Geophys. Res., 108(D13), 8471, doi:10.1029/2002JD002250, 2003.

Johnson, B. T., B. Heese, S. A. McFarlane, P. Chazette, A. Jones, and N. Bellouin (2008), Vertical distribution and radiative effects of mineral dust and biomass burning aerosol over West Africa during DABEX, J. Geophys. Res., 113, D00C12, doi:10.1029/2008JD009848.

Magi, B. I., Q. Fu, and J. Redemann (2007), A methodology to retrieve self-consistent aerosol optical properties using common aircraft measurements, J. Geophys. Res., 112, D24S12, doi:10.1029/2006JD008312.

Osborne S.R., J.M. Haywood Aircraft observations of the microphysical and optical properties of major aerosol species Atmospheric Research 73 (2005) 173–201.

Schmid, B., et al., Coordinated airborne, spaceborne, and ground-based measurements of massive thick aerosol layers during the dry season in southern Africa, J. Geophys. Res., 108(D13), 8496, doi:10.1029/2002JD002297, 2003.

Anderson, T. L., S. J. Masonis, D. S. Covert, N. C. Ahlquist, S. G. Howell, A. D. Clarke, and C. S. McNaughton, Variability of aerosol optical properties derived from in situ aircraft measurements during ACE-Asia, J. Geophys. Res., 108(D23), 8647, doi:10.1029/2002JD003247, 2003.

Carrico, C. M., P. Kus, M. J. Rood, P. K. Quinn, and T. S. Bates, Mixtures of pollution, dust, sea salt, and volcanic aerosol during ACE-Asia: Radiative properties as a function of relative humidity, J. Geophys. Res., 108(D23), 8650, doi:10.1029/2003JD003405, 2003.

Redemann, J., S. J. Masonis, B. Schmid, T. L. Anderson, P. B. Russell, J. M. Livingston, O. Dubovik, and A. D. Clarke, Clear-column closure studies of aerosols and water vapor aboard the NCAR C-130 during ACE-Asia, 2001, J. Geophys. Res., 108(D23), 8655, doi:10.1029/2003JD003442, 2003.

Howell, S. G., A. D. Clarke, Y. Shinozuka, V. Kapustin, C. S. McNaughton, B. J. Huebert, S. J. Doherty, and T. L. Anderson (2006), Influence of relative humidity upon pollution and dust during ACE-Asia: Size distributions and implications for optical properties, J. Geophys. Res., 111, D06205, doi:10.1029/2004JD005759.

Andrews E., P. J. Sheridan, and J. A. Ogren Seasonal differences in the vertical profiles of aerosol optical properties over rural Oklahoma Atmos. Chem. Phys., 11, 10661–

[Figure]

Esteve A. R, J. A. Ogren, P. J. Sheridan, E. Andrews, B. N. Holben, and M. P. Utrillas Sources of discrepancy between aerosol optical depth obtained fromAERONET and in-situ aircraft profiles Atmos. Chem. Phys., 12, 2987–3003, 2012

2. It is known that data from in situ measurements of the absorption and scattering coefficients are characterized by relatively high degree of uncertainty. I think it is advisable to consider the influence of these factors when carrying out comparisons (including PSDs).

3. The authors compare in situ extinction coefficient and the average aerosol extinction as AOD contained in ML : $\sigma$Sun ptotometer=AODSun photometer/MLH. Why do the authors have chosen namely this characteristic? After filtering of the multilayer cases the extinction coefficient decreases with height. It can be assumed that the decrease is described by an exponential or linear law. It is possible that for this class of atmospheric situations can be used not average value and it will be more physically substantiated.

4. It seems to me that it is necessary to write more clearly what atmospheric situations belong to the ML cases and what – to FT cĐřses.

5. Page 5: "A CIMEL Sun photometer (CE-318), operating at the CZ site, measures the aerosol optical properties of the total integrated atmospheric column under ambient conditions at four wavelengths (440, 675, 870 and 1020 nm)". The measurements of the diffuse radiation at these wavelengths provide a solution of the inverse problem (retrieval of phase scattering function and single scattering albedo, refractive index, particle size distribution function). AOD is also the optical characteristic, but AOD measurements is performed on the extended set of wavelength.

The way of presenting the results in this version of the manuscript has largely descriptive character (5% more than ..., the correlation coefficient is equal to ...). I think the text needs refinement: the article will be more interesting if it will be supplemented by

an more detailed analysis of the causes that led to the presented above results.

Please also note the supplement to this comment:
http://www.atmos-meas-tech-discuss.net/amt-2016-97/amt-2016-97-RC2-supplement.pdf

---

## Author Comment (AC1) · 31 Aug 2016

First, authors would like to thanks both reviewers to their time and to bring interesting comments on scientific contributions to the paper. We answer below point by point to these comments and to raised questions:

RC1 : A set of papers presents analysis of correlation between parameters of aerosol particles received from in-situ measurements at the ground-based sites or at meteorological towers and the results of columnar or altitude- resolved remote sensing data. Nevertheless, this multiple-factor problem remains relevant for interpreting data of com-

[Figure]

plex atmospheric experiments. The manuscript by Chauvigné et al presents novel results for characterization of this problem because of uniqueness of the database gathered at the PUY atmospheric station. The authors analyze the results of long-term in-situ meteorological, aerosol optical and microphysical observations at two sites with altitude 0.4 and 1.4 km, as well as data of lidar and sun-radiometer measurements. Depending on meteorological conditions, the upper site of in-situ measurements was located either within mixing layer, or in free troposphere which allows the different variants of aerosol investigations to be considered. Paper can be recommended for publication. List of specific comments:

Page 8, Line 18, 19: Remove "Error! Reference source not found" and correct the text.

Authors :

The Figure has been removed and thus the referencing text.

Discussion, page 9, line 14: Reference removed.

–

RC1 : Page 8, Line 26: It is appropriate to note that results of in-situ measurements in the ground-based layer are affected by local aerosol sources. Measurement conditions at the high altitude Puy de Dôme site eliminate such interferences.

Authors :

Indeed, the high altitude site is not affected by local pollution, while it might have been the case at the ground-based site in the study by Bergin et al., (2000). We modified the sentence:

Discussion, page 9, line 23: "This result might be explained by a better representativeness of the atmospheric column by high altitude in-situ measurements, that are usually representative of a large spatial area (Henne et al., 2010) and less affected by eventual local contaminations than ground-based low altitude sites might be. "

–

RC1 : Figure 7 and 8: what means Diameter? (mode, median . . ..of fine/coarse particle volume distribution ?

Authors :

These are diameters obtained after the fitting procedure of the volume size distribution: we mean modal diameters.

Figure 7 legend: "Scatter plot of fine and coarse mode modal diameters, and volume concentrations after applying the ML and FT contribution factors to Sun Photometer concentrations. Blue fit and markers are for FT data and red fit and markers for ML data."

Figure 8 legend: "Scatter plot of fine and coarse mode modal diameters, and volume concentrations after applying the Humidity growth factor (HGF). Color corresponds to the mean relative humidity between the two stations. Blue fit and cross markers are for FT data and red fit and circle markers for ML data."

–

RC1 : Part 5.2 The significant difference between "in-situ and Sun-photometer diameters" of fine particles is a very interesting result. Do authors have a physical interpretation of this feature?

Authors :

Difference of particle diameters between the two measurement techniques can be explain mainly by a vertical gradient of aerosol diameters within the atmosphere and by different vertical transport processes.

Discussion, page 13, line 13: "However, a vertical gradient of aerosol diameters might explain this feature. Liu et al. (2009) have analyzed higher effective radii at high altitude than at ground level from an aircraft study above Beijing region. Indeed, large

aerosols are transported more efficiently over large distances when they are transported at higher wind speeds, which are prevailing at high altitudes."

---

## Author Comment (AC2) · 31 Aug 2016

First, authors would like to thanks both reviewers to their time and to bring interesting comments on scientific contributions to the paper. We answer below point by point to these comments and to raised questions:

1. RC2 : The paper reports the results of the comparison of the aerosol optical and microstructural characteristics in the atmospheric column and at altitude where the PUY site is located. At PUY site aerosol light absorption ($\sigma$abs) and scattering ($\sigma$scat) coefficients are measured using a Multi Angles Absorption photometer

[Figure]

and a three wavelengths nephelometer respectively. Similar equipment is installed on the aircraft to provide vertical profiles $\sigma$abs(z), $\sigma$scat(z) and extinction coefficient $\sigma$ext(z)=$\sigma$abs(z)+$\sigma$scat(z). The analysis of measurements of aerosol characteristics in the atmospheric column and their vertical profiles obtained on board of aircraft is presented in a number of studies (see **). Several papers also discusses the effect of relative humidity on the absorption $\sigma$abs(z) and scattering $\sigma$scat(z) coefficients. This manuscript does not consider the vertical profiles of the extinction coefficient, however, some approaches and results (** and others) logical to use in current study. Is possible it will be useful in the interpretation of data and help to explain the significant difference between in-situ and Sun-photometer measurements. The above also applies to the particle size distribution function.

Authors:

Yes, we agree that aircraft aerosol measurements along the atmospheric column are very informative for comparison with remote sensing measurements. We already mentioned a few airborne studies, and the order of magnitude of the discrepancies between in situ vertical distribution and remote sensing measurements. We now precise these discrepancies using the additional reference list given by reviewer #2 in the introduction:

Discussion, page 3-4, line 28 : Âń Several other studies use aircraft in-situ measurements in order to describe the entire atmospheric column during specific research campaigns (ACE-Asia, ACE-2, SAFARI 2000,. . .) in comparison to remote sensing measurements performed at the same time. General results agree that aircraft vertical profiles are well correlated to Sun photometer measurements (Haywood et al., 2003 ; Magi et al., 2007 ; Johnson et al., 2008). However, some features can influence the correlation. Haywood et al. (2003) show that aircraft pitch can influence in-situ measurements and data need to be corrected either during the ascent or the descent. Despite a good correlation between in-situ and Sun photometer AOD measurements, authors highlight that in-situ measurements overestimate extinction measurements (around 25%) in opposition to Müller et al. (2012) and Schmid et al. (2003) works. Magi et al, (2007) indicate that the particle shape and wavelength measurement can significantly influence both remote sensing and in-situ measurements. Johnson et al. (2008) report good correlation but an important factor (up to a factor 2 in absolute value) between ground base remote sensing and in-situ aircraft measurements mainly due to distance between the two measurement techniques (up to 100 km). On average, authors report aircraft AOD 30% under AERONET measurements at 550 nm, also shown by Osborne et al. (2008). The particle size distributions are well correlated between Sun photometer and aircraft measurements between 0,05 and 1 $\mu$m (Haywood et al., 2003). However, the particle size distribution can also be biased by vertical heterogeneity of the atmosphere and measurements synchronization between different sampling methods (Osborne and Haywood, 2005). The particle hygroscopicity also has an important influence on aerosol properties and hence on the comparison between measurement techniques. Carrico et al. (2003) highlight the impact of ambient relative humidity on aerosol optical properties for different aerosol natures and from aircraft measurement during ACE-Asia experiment. The ratio between ambient scattering property and the dry measurement (RH=19%) from in-situ probes vary from 1 to 1.6 from dust to marine aerosols, which is of the order of magnitude of the discrepancy between in situ and remote sensing measurements in the literature. Âż

We also use these new references to compare with our results:

Discussion, page 11, line 12 : Âń Correlations under ML conditions show very close similarities to some previous studies (30% discrepancy reported from Johnson et al. (2008) and Osborne et al. (2008) from aircraft measurements during DABEX experiments).Âż

Discussion, page 12, line 26 : Âń Previous works show some impact of sampling techniques on the modal standard deviation (Osborne and Haywood, 2005) leading to higher integrated concentrations from Sun photometer measurements. In this particular work, the distance between the two sites is short enough to limit horizontal

inhomogeneities.Âż

Discussion, page 14, line 19 : Âń Aerosols may be more hygroscopic than predicted by the parameterization if they are sea salt aerosols, or less hygroscopic than predicted if they are Saharan dust (Carrico et al., 2003). Âż

–

2. RC2 : It is known that data from in situ measurements of the absorption and scattering coefficients are characterized by relatively high degree of uncertainty. I think it is advisable to consider the influence of these factors when carrying out comparisons (including PSDs).

Authors:

We now mention the uncertainties of in-situ probes previously documented in the literature.

Discussion, page 5, line 21 : Âń Both instruments reveal uncertainties in number concentrations around 10% (Venzac et al., 2009 : Burkart et al., 2010) mainly due to the flowrate, the cut-off diameters and the CPC counting efficiency. Âż

Discussion, page 6, line 6 : Âń Uncertainties were previously reported for nephelometer and MAAP measurements were in the range of 1 to 5% due to truncation and angular non idealities in the light source (Anderson et al., 1996 and Bond et al., 2009) and around 12% for MAAP measurements (Petzold and Schönlinner (2004). Âż

Discussion, page 9, line 21: "The small dispersion of the comparison can be due to instruments uncertainties estimate around 5% for in-situ measurements (Bond et al., 2009) and 0.02 for Sun photometer AOD (Dubovik et King, 2000)."

Discussion, page 13, line 26: "This effect may also be partly explained by uncertainties from in-situ probes (around 10%) (Venzac et al., 2009 : Burkart et al., 2010) on one hand, and uncertainties from sun photometer measurements on the other hand (be-

tween 15 and 35% between 0.1 $\mu$m and 7 $\mu$m and between 30 and 100% out of this range) (Dubovik et al. (2000)))."

–

3. RC2 : The authors compare in situ extinction coefficient and the average aerosol extinction as AOD contained in ML : $\sigma$Sun ptotometer=AODSun photometer/MLH. Why do the authors have chosen namely this characteristic? After filtering of the multilayer cases the extinction coefficient decreases with height. It can be assumed that the decrease is described by an exponential or linear law. It is possible that for this class of atmospheric situations can be used not average value and it will be more physically substantiated.

Authors :

It is actually one of the main conclusion of our work that the ML seems to not be fully well mixed and that a significant vertical gradient exists, as already reported by (Andrews et al., 2011). However, several recent studies use the assumption of an homogeneous mixing layer and others observe dry extinction roughly independent of altitude (Kanitz et al., 2013 ; Wagner et al., 2015). The Sheridan et al. (2012) study, based on several aerosol profile measurements (aircraft in-situ, ground based remote sensing and from space), show that the ML can be well mixed. This highlights the necessity to test this assumption. Our strategy was to first start with very simple hypothesises (all aerosol contained in the ML, or two layers homogeneous but distinct) and evaluate how these hypothesis impact on the comparisons. However, we agree that a linearly decreasing concentration within the whole atmosphere is one additional view of the atmosphere that we could test. This was performed on the extinction coefficient as an additional figure (Figure 4c)). The method is described in the text. Results show that this hypothesis decreases the distance to the line 1:1 compare to Figure 4b) (from 85% to 10%) and increase the correlation coefficient between methods (from 0.73 to 0.82). Hence we conclude that 1. The major part of the aerosol is not contained in the ML

and 2. Linear decrease of the aerosol concentrations can be one solution to improve vertical aerosol profiles.

This result have been added to the text and to the Figure 4 c):

Abstract: "Moreover, the assumption of a decreasing linear vertical aerosol profile in the whole atmosphere has been tested, significantly improving the instrumental agreement."

Discussion, page 10, line 12: "The assumption of homogeneous ML can also explain the significant difference between in-situ and remote sensing measurements. A linear decreasing profile of the aerosol extinction with altitude has been calculated using two constrains: - The extinction coefficient is equal to zero at 5 km high. - The integration of vertical extinction is equal to the Sun photometer AOD. The Sun photometer extinction is then retrieved from the linear equation at 1465m. Results show better agreement between the two measurement techniques (Figure 4c)). However, several recent studies use the assumption of an homogeneous mixing layer and others observe dry extinction roughly independent of altitude (Kanitz et al., 2013 ; Wagner et al., 2015). The Sheridan et al. (2012) study, based on several aerosol profile measurements (aircraft in-situ, ground based remote sensing and from space), show that the ML can be well mixed. Hence, we also test the hypothesis of a constant aerosol profile within the ML."

Discussion, page 12, line 6: "In comparison to linear aerosol vertical decreasing model (Figure 4 c)), the present results are still less correlated for both layers. Despite a number of studies using homogeneous mixing layer model (Sheridan et al., 2012 ; Kanitz et al., 2013 ; Wagner et al., 2015), the assumption to use homogeneous decreasing of aerosol concentration from the ground to 5000 m a.s.l. seems to be in better agreement with the integrated Sun photometer measurements. However, as mentioned by Andrews et al. (2011) study, a separate definition between ML and FT layers will be more appropriate."

Conclusion, page 15, line 5: "A linear decrease of aerosol extinction with altitude across

the whole tropospheric column permits to obtain a better agreement between the two measurement techniques. Our next step was to test the assumption of two different distinct layers (ML and FT) in which the aerosol concentrations are homogeneously distributed. "

Conclusion, page 15, line 15: "The atmospheric profile model assuming linear decreasing of aerosol concentrations with altitude give interesting results in the comparison of in-situ and remote sensing measurements. Taking into account this decreasing profile up to 5000 m a.s.l. (thus including both the ML and the FT) leads to the best correlation between extinction coefficients measured by the two techniques."

Conclusion, page 15, line 32: "The study also report interesting alternative to use linear aerosol vertical decreasing model for the comparison."

–

4. RC2 : It seems to me that it is necessary to write more clearly what atmospheric situations belong to the ML cases and what – to FT cĐřses.

Authors :

Both ML and FT properties have been studied from PUY station measurements and identified through aerosol concentrations (Venzac et al., 2009) clearly higher within the mixing layer. Clarifications have been added to the text.

Discussion, page 7, line 28: "The diurnal variation of the mixing layer is driven by surface heating. Depending of the time of the day and seasons, the mixing layer limit can be above (ML cases) or below (FT cases) the PUY station altitude (1465 m a.s.l.). The PUY station was calculated using the ECMWF model to be in the ML usually during daytime and the warm seasons (spring and summer) and in the FT during nightime during the cold seasons (autumn and winter) (Venzac et al., 2009). These variations were confirmed in the present study when using the LIDAR profiles."

Discussion, page 11, line 17: "When the PUY site is in the ML (WCT > 1200 m) [. . .]

when the Puy de Dôme in-situ measurements are performed in the FT (WCT < 1200 m)."

–

5. RC2 : Page 5: "A CIMEL Sun photometer (CE-318), operating at the CZ site, measures the aerosol optical properties of the total integrated atmospheric column under ambient conditions at four wavelengths (440, 675, 870 and 1020 nm)". The measurements of the diffuse radiation at these wavelengths provide a solution of the inverse problem (retrieval of phase scattering function and single scattering albedo, refractive index, particle size distribution function). AOD is also the optical characteristic, but AOD measurements is performed on the extended set of wavelength.

Authors :

Descriptive report of CIMEL instruments specifies the possibility to retrieve AOD and inversed retrievals at more than four wavelengths depending on the instrument model. Details has been added to the text.

Discussion, page 6, line 16: "Due to Dubovik's algorithm, AOD and inversed properties can be retrieved at the same wavelengths. Depending on the instrument, the measurements can be taken on all or some of the following channels: 340, 380, 440, 500, 675, 870, 1020 and 1640 nm."

–

RC2 : The way of presenting the results in this version of the manuscript has largely descriptive character (5% more than . . ., the correlation coefficient is equal to ...). I think the text needs refinement: the article will be more interesting if it will be supplemented by an more detailed analysis of the causes that led to the presented above results.

Authors :

We believe that taking into account some suggestions of the reviewer helped to better

highlight the causes of discrepancies between the in situ and sun photometer measurements. However, we choose to submit this paper to a "technical" journal and did not aim at using the results to conclude to fundamental scientific conclusions regarding the structure of the atmosphere. Our goal is to provide estimates of the order of magnitude of the discrepancy between two very different measurement techniques, and the conditions associated to these results.
* * *